# STAT3 Activation and Oncogenesis in Lymphoma

**DOI:** 10.3390/cancers12010019

**Published:** 2019-12-19

**Authors:** Fen Zhu, Kevin Boyang Wang, Lixin Rui

**Affiliations:** 1Department of Medicine, University of Wisconsin School of Medicine and Public Health, Madison, WI 53792, USA; fzhu29@wisc.edu; 2Carbone Cancer Center, University of Wisconsin School of Medicine and Public Health, Madison, WI 53792, USA; 3Department of Orthopaedic Surgery, Southeast Regional Hospital, Bega, NSW 2550, Australia; dr.kevinb.wang@gmail.com

**Keywords:** STAT3, lymphoma, oncogene, gene transcription and targeted therapy

## Abstract

Signal transducer and activator of transcription 3 (STAT3) is an important and the most studied transcription factor in the Janus kinase (JAK)/STAT signaling pathway. STAT3 mediates the expression of various genes that play a critical role in many cellular and biological processes, such as cell proliferation, survival, differentiation, migration, angiogenesis, and inflammation. STAT3 and associated JAKs are activated and tightly regulated by a variety of cytokines and growth factors and their receptors in normal immune responses. However, abnormal expression of STAT3 leads to its constitutive activation, which promotes malignant transformation and tumor progression through oncogenic gene expression in numerous human cancers. Human lymphoma is a heterogeneous malignancy of T and B lymphocytes. Constitutive signaling by STAT3 is an oncogenic driver in several types of B-cell lymphoma and most of T-cell lymphomas. Aberrant STAT3 activation can also induce inappropriate expression of genes involved in tumor immune evasion such as *PD-L1*. In this review, we focus on the oncogenic role of STAT3 in human lymphoma and highlight potential therapeutic intervention by targeting JAK/STAT3 signaling.

## 1. Introduction

Signal transducer and activator of transcription 3 (STAT3) was discovered in 1993 by a biochemical study, in which STAT3 functions as an acute phase response factor and binds to its consensus DNA motif in nuclear extracts from the liver cell line HepG2 upon stimulation with the cytokine IL-6 [1]. In the following year, STAT3 was molecularly cloned and further characterized as a signal transduction molecule that can be phosphorylated by signaling from receptors of the IL-6 family cytokines [2,3,4]. Since then, STAT3 has been the most studied member of the STAT family, with a total of more than 20,000 publications based on the PubMed database. In addition to the IL-6 family cytokines, STAT3 can be activated by many other cytokines, such as the common γ chain (γc) receptor family and various growth factors as well as nonreceptor tyrosine kinases [5,6]. When a cytokine binds to its cognate receptor, the receptor is dimerized and phosphorylated on its intracellular tail. These receptor activation events induce cross-Janus kinase (JAK) tyrosine phosphorylation and create docking sites for STAT3, where the activated JAK phosphorylates STAT3 [5,6]. Phosphorylation of STAT3 on Tyr-705 in the Src homology 2 (SH2) domain facilitates STAT3 dimerization and nuclear translocation, where phosphorylated dimers directly bind to DNA consensus sequences for gene transcription [5,6]. STAT3 can recruit nuclear co-activators, possibly through a transactivation (TAD) domain, to enhance its transcriptional activity [5,6].

STAT3, a key transcription factor in the JAK/STAT signaling pathway, regulates expression of genes that control various cellular and biological processes, including cell proliferation, survival, differentiation, migration, angiogenesis, inflammation, and immune responses [7,8]. STAT3 activation is tightly controlled and only occurs within a short window of time during normal immune responses [8]. Accumulating evidence, however, demonstrates dysregulation and constitutive activation of STAT3 that are associated with several human diseases including cancer [5,8]. The mechanisms underlying aberrant STAT3 activation during tumor development include autocrine action of cytokines, cytokine receptor amplification and activating mutations, gain-of-function mutations in STAT3 and JAKs, and loss of function mutations in negative regulators (e.g., Protein Tyrosine Phosphatase Receptor Type T (PTPRT), SOCS1, and SOCS3), depending on the types an the stages of individual cancers [5,6,9,10,11]. In this review, we summarize the mechanisms of STAT3 activation and oncogenesis in B- and T-cell lymphomas and discuss potential therapeutic intervention. 

## 2. STAT3 Domains and Alternative Splicing Isoforms

STAT3 and other STAT proteins contain six domains: a conserved amino-terminus, a coiled-coil domain, a DNA-binding domain (DBD), a linker domain, the SH2 domain, and a carboxy-terminal TAD (Figure 1) [12]. The SH2 domain is a key functional domain, which mediates STAT3 transit binding to cytokine receptors, STAT3 dimerization, and consequent nuclear translocation [12]. This action is modulated by phosphorylation events close to the C-terminus of STAT3 [13]. Phosphorylation of Tyr-705 in the SH2 domain is well characterized as an early event for STAT3 activation, which is mediated by JAKs and nonreceptor kinases, such as Src and Abl [5,14] autoimmune diseasesions in lymphoma cells. AT3b through a high concerved acceptor site in exon 23, main for receptor binding a STAT3 transcriptional activity is enhanced by phosphorylation of serine (Ser)-727 on the TAD [15,16]. The underlying mechanism is less understood, but the serine phosphorylation can facilitate recruitment of co-activators to the STAT3 transcription site [17]. In addition, Thr-714 phosphorylation together with Ser-727 phosphorylation is required for STAT3 transcriptional activity and the serine/threonine kinases GSK3α and phosphorylate both Ser-727 and Thr-714 [18].

The *STAT3* gene has two splice sites close to its 3′ end, which create four STAT3 splice variants, termed α, β, ΔS-α, and ΔS-β (Figure 1) [19,20]. A highly conserved acceptor site in exon 23 generates either STAT3 α that includes a 55-residue TAD or STAT3β, which contains seven unique residues at the C-terminus but lacks the TAD [19,20]. A short-range donor (5′) splice site results in ΔS-α and ΔS-β, in which three nucleotides that encode Ser-701 are excluded (Figure 1) [20,21]. We were able to detect expression of all four isoforms of STAT3 in eosinophils and diffuse large B-cell lymphoma (DLBCL) cells by qPCR analysis [20]. We found that the expression pattern is similar between the two types of cells. STAT3α is the most abundant isoform, accounting for ~75% [20]. STAT3β is expressed at ~10%, and STAT3ΔS-α and STAT3ΔS-β together contribute to the remaining ~15% of expression [20].

STAT3α plays a major role in STAT3-associated oncogenesis [12]. Expression of constitutively active STAT3α promotes proliferation, tumor growth, and metastasis in various tumor models [12,22,23]. An early elegant genetic study has demonstrated that activating mutations (A661C and N663C) within the C-terminal SH2 domain facilitate STAT3α dimerization, nuclear translocation, and gene transcription, leading to cellular transformation [24]. Phosphorylation of STAT3 on Ser-727 by protein kinase C in solid tumor cell lines leads to STAT3α activation, which induces expression of genes involved in cancer migration and invasion [25]. STAT3α is upregulated in acute myeloid leukemia (AML) cells when stimulated with G-CSF, which may enhance proliferation and block differentiation [26]. STAT3β has been suggested as a negative regulator of STAT3α and a tumor suppressor despite its functional enigma [12]. For example, STAT3β expression inhibits the survival and proliferation of human ovarian and breast cancer cells, in which STAT3 is constitutively activated [27,28]. Overexpression of STAT3β in B16 melanoma cells diminishes STAT3 DNA-binding activity, blocks STAT3-mediated gene expression and reduces cell proliferation [29]. The same group has also demonstrated antitumor effects of STAT3β in vivo [30]. The tumor suppressor function of STAT3β is further demonstrated in xenograft mouse models [31,32] and by a recent AML study [33].

STAT3ΔS-α and -ΔS-β are less studied, and biology of these two isoforms with a deletion of serine 701 is unknown. STAT3 is constitutively activated in the subtype of activated B-cell-like (ABC) DLBCL cells [34]. Constitutive activation of STAT3 is required for the survival and proliferation of ABC DLBCL cells [34,35]. We examined the function of STAT3 isoforms in ABC DLBCL using a knockdown/re-expression strategy and tested whether expression of STAT3ΔS-α or -ΔS-β is required for the survival of DLBCL cells [36]. We used a constitutively activated form of STAT3 (STAT3-C) as a positive control [24]. The results revealed that the best rescue effect was observed when all four isoforms were re-expressed [36]. A partial rescue was achieved with re-expression of STAT3α and STAT3ΔS-α or STAT3β and STAT3ΔS-β in pairs [36]. Re-expressing any of the four isoforms had very limited rescue effects [36]. These results suggest that expression of STAT3ΔS-α and -ΔS-β is not dispensable for optimal STAT3 function in ABC DLBCL cells.

## 3. The Role of STAT3 in Human Lymphoma

### 3.1. STAT3 Mutations

Somatic STAT3 mutations are largely restricted to hematological disorders (Table 1). STAT3 mutations have been discovered in 40% of patients with T-cell large granular lymphocytic leukemia (T-LGL), a cytotoxic T-cell malignancy that is often associated with autoimmune diseases [38]. Highly frequent mutations of STAT3 are also present in large granular lymphocytic leukemia of natural killer cells (NK-LGL) (30%) [37]. STAT3 is mutated in 6% of natural killer/T-cell lymphomas (NKTCL) and 8% of peripheral T-cell lymphoma (PTCL) [43]. STAT3 mutations are rare in B-cell non-Hodgkin lymphoma and other T-cell malignancies [52,57,60]. Hot spots of STAT3 missense mutations are located in the SH2 domain, including Y640F, D661V, and D661Y (Table 1) [7]. STAT3 gain-of-function mutations have been assessed by an increase in its phosphorylation and transcriptional activity, which can promote the survival and proliferation of malignant cells (Table 1).

### 3.2. Diffuse Large B-Cell Lymphoma

DLBCL is a type of aggressive non-Hodgkin lymphoma, accounting for 30–40% of newly diagnosed cases [62]. Gene expression profiling has identified two main molecular subtypes termed activated B-cell-like (ABC) and germinal center B-cell-like (GCB) DLBCL cells [63,64,65]. ABC DLBCL cells bear genetic alterations in the Toll-like receptor (TLR) and B-cell receptor (BCR) signaling pathways [66,67,68]. These two hallmark oncogenic pathways lead to high NF-κB activity [69] that induces production of IL-6 and IL-10 in ABC DLBCL cells (Figure 2) [68]. Autocrine action of IL-6 and IL-10 results in constitutive activation of JAK1 [70] and STAT3 [34,36,71,72], which promotes the survival and proliferation of ABC DLBCL cells. In addition to a canonical pathway, we characterized an epigenetic mechanism of JAK1 in gene regulation [70,73,74]. JAK1 is transferred into the nucleus, where the kinase directly phosphorylates histone H3 to activate gene expression [70]. We identified ~3000 target genes by JAK1 through this mechanism, including NF-κB pathway genes [70]. These findings demonstrate a feed-forward regulatory mechanism between the JAK1 and NF-κB signaling pathways in gene expression, which may play an important role in the pathogenesis of ABC DLBCL.

In addition to promoting cell survival, STAT3 activation has been found to be associated with an inferior clinical outcome in DLBCL [75,76]. To investigate the molecular mechanisms underlying STAT3 oncogenesis in ABC DLBCL, we recently identified ~2000 STAT3 target genes in ABC DLBCL by genomic analysis [35]. These include NF-κB pathway genes (e.g., *NFKB2*, *NFKBIA*, *NFKBIZ*, *CXCL13*, and *TRAF1*), JAK/STAT pathway genes (e.g., *IL-10*, *SOCS3*, and *STAT3*), antiapoptotic pathway genes (e.g., *BCL2*, *MCL1*, *BCL2L11*, and *CASP8*), and PI3K/AKT/mTORC1 pathway genes (e.g., *AKT2*, *SCL7A11*, and *ACSL3*) (Figure 2) [35,77]. In addition, STAT3 functions as a transcriptional suppressor and inhibits transcription of genes in the type I interferon signaling pathway, including *IRF7*, *IRF9*, *STAT1*, and *STAT2* [35,77]. Anti-type I interferon properties of STAT3 have been characterized as an antiviral defense mechanism [78,79,80]. The type I interferon pathway is lethal to ABC DLBCL cells, which can be also suppressed by the transcription complex IRF4/SPIB [81].

We also performed genome-wide analysis in both ABC DLBCL and normal activated B-cells and identified 1014 STAT3 targets that are specific for ABC DLBCL, including the immune checkpoint molecule PD-L1, the transcriptional activator MEF2B, the hematopoietic cell kinase (HCK), the Epstein–Barr virus-induced genes 3 (*EBI3*), and genes involved in regulating cellular metabolism (e.g., *SCARB1*) [77]. PD-L1 is a well-studied immune checkpoint molecule that binds to the inhibitory PD-1 receptor on T-cells and suppresses T-cell activation [82]. In DLBCL, PD-L1 overexpression has been found in ~25% of the ABC subtype but is rare in the GCB subtype [83]. High levels of PD-L1 expression are correlated with a poor survival outcome in DLBCL [83,84]. Several mechanisms underlying PD-L1 overexpression have been investigated, including genomic amplifications and translocations [85] and BCR-mediated NFATc1 activation through IL-10/STAT3 [86].

In conclusion, the JAK1/STAT3 signaling pathway plays an important role in the pathogenesis of DLBCL. Constitutively active STAT3 regulates expression of genes that are involved in multiple oncogenic processes in ABC DLBCL, including promoting cell survival, proliferation, and migration, reprogramming metabolism and facilitating tumor immune evasion. Targeting JAK1/STAT3 signaling represents a promising therapeutic strategy in ABC DLBCL.

### 3.3. Mantle Cell Lymphoma

STAT3 is phosphorylated in 20–40% of mantle cell lymphoma (MCL), an aggressive, rare form of B-cell non-Hodgkin lymphoma [87,88]. Constitutive activation of STAT3 can result from IL-6, which is produced by MCL cells or cells in tumor microenvironment [89]. SOCS3, a negatively regulator of the JAK–STAT signaling pathway, loses its expression in ~50% of MCL due to DNA methylation [90]. A recent study has revealed a unique role of STAT3 in MCL, which demonstrates that STAT3 represses expression of SOX11, a critical transcription factor that promotes lymphomagenesis in MCL [91]. This finding indicates that STAT3 may not be an ideal target in MCL given its negative role in SOX11 regulation. 

### 3.4. Hodgkin Lymphoma and Primary Mediastinal B-Cell Lymphoma

Classical Hodgkin lymphoma (HL), a common cancer in young adults, is characterized by the presence of Reed-Sternberg cells, which harbor various recurrent genetic lesions, including chromosome 9p24 amplification (e.g., copy number gain of JAK2, PD-L1, and PD-L2) and mutations in genes in the JAK–STAT and NF-κB signaling pathways (e.g., STAT6, SOCS1, TNFAIP3, and NFKBIE) [92,93,94,95,96,97]. HL shares genetic and biological features and oncogenic mechanisms with primary mediastinal B-cell lymphoma, a subtype of DLBCL [94,98]. These genetic alterations result in constitutive activations of the antiapoptotic and proinflammatory NF-κB and JAK–STAT signaling pathways, as well as tumor immune evasion [95,96]. STAT3 phosphorylation has been found in HL cell lines and primary Reed-Sternberg cells [99,100]. Inhibition of JAK2/STAT3 activation can trigger cell apoptosis in vitro [101] and inhibit tumor growth in an HL xenograft mouse model [102]. In addition to the genomic amplification, PD-L1 and PD-L2 can be upregulated by the co-amplified gene *JAK2* in primary mediastinal B-cell lymphoma and HL [94]. Targeting the PD-1/PD-L pathway can significantly improve progression-free survival time in 86% of patients with relapsed or treatment refractory HL [103].

### 3.5. Multiple Myeloma

Multiple myeloma (MM) is a cancer of plasma cells (terminally differentiated B-cells) in the bone marrow, accounting for 10% of all hematological malignances [104]. Hallmark oncogenic signaling pathways in MM include JAK-STAT3, PI3K/AKT/mTOR, and NF-κB [105,106], in which STAT3 is activated by the IL-6–JAK–STAT3 axis [107,108]. A positive feedback regulatory loop between STAT3 and phosphatase of regenerating liver 3 (PRL-3) is another mechanism underlying STAT3 activation in MM [107]. Hypermethylation and silencing of the negative regulators SOCS1 and SHP1 can also contribute to high STAT3 activity in MM [109,110]. Expression of antiapoptotic pathway genes such as *MCL-1*, *BCL2*, and *BCL-XL* due to STAT3 activation promotes MM cell survival [107,111]. STAT3 activity promotes expansion and activation of myeloid-derived suppressor cells (MDSCs), which mediate immunosuppression and facilitate tumor progression in the MM bone marrow microenvironment [112,113,114]. Expression of phosphorylated STAT3 is associated with poor prognosis and survival in MM patients [115]. Several preclinical studies have demonstrated that inhibition of JAK/STAT3 signaling alone or in combination with other antitumor reagents inhibits MM cell growth both in vitro and in vivo [116,117,118,119].

### 3.6. T-Cell Lymphomas

The JAK/STAT signaling pathway mediated by the common γc receptor-dependent cytokines IL-2, -4, -7, -9, -15, and -21 plays critical roles in T-cell immune responses [120]. Abnormal activation of the JAK/STAT pathway, which can be assessed by STAT3 and STAT5 phosphorylation, is pervasive in diverse T-cell malignancies [121]. In peripheral T-cell lymphoma (PTCL) and extranodal NKTCL, recurrent mutations in STAT3 are most frequent, followed by JAK1, JAK3, and SOCS1 (Table 1) [44]. STAT3 mutations are activated and lead to increased phosphorylation and transcriptional activity of STAT3 [25]. In addition to promoting cell survival and proliferation, constitutive activation of STAT3 in cancer cells causes high levels of PD-L1 expression, which may facilitate tumor immune evasion (Figure 3) [44]. Activating STAT3 mutations are present in 50% of NK cell lines and 67% of γδ-T-cell lines (67%) but only in 5.9% of NKTCL and 8.3% of γδ-PTCL patient samples [43]. This suggests that these mutations are critical in cell survival independent of stromal components and therefore selectively enriched in cell lines. In anaplastic large cell lymphoma (ALCL), JAK1 and STAT3 are activated by high levels of cytokines, including IL-6 [122]. The treatment with the JAK1/2 inhibitor Ruxolitinib reduces xenografted ALCL tumor growth [122].

## 4. Targeting JAK/STAT3 Signaling in Lymphoma

### 4.1. Direct STAT3 Inhibition

Aberrant expression and constitutive activation in several types of B-cell lymphoma and in almost all T-cell lymphomas render STAT3 as a promising potential therapeutic target. STAT3 inhibitors have been developed to interact with the STAT3 domains, with a major focus on the SH2 domain and the DBD [123]. The STAT three inhibitor compound (Stattic) is an SH2-domain inhibitor, discovered by a high-throughput chemical library screen [124]. Stattic inhibits STAT3 activation by binding to cysteine residues close to the phosphopeptide-binding area of the SH2 domain [124]. Other recently developed inhibitors targeting the SH2 domain of STAT3 include S3I-201 [125], its analog S3I-1757 [126], STA-21, and its derivatives LLL-3 and LLL-12 [127,128,129]. Effectiveness of these inhibitors in STAT3 inhibition has been tested in solid cancers [128,130,131,132]. The small peptide aptamer DBD-1 is an STAT3 DBD inhibitor, which prevents STAT3 from DNA binding but does not affect STAT3 activation [133]. InS3-54 is another DBD inhibitor, which demonstrates selectivity for STAT3 over STAT1 [134,135]. STAT3 SH2-domain inhibitors are slow to move into clinical medicine due to high homology of the SH2 domain between STAT3 and other family members as well as high concentrations required for disruption of protein–protein interactions, which increase off-target toxicities [123].

To date, no STAT3 inhibitors have yet been approved to treat cancer by the Food and Drug Administration (FDA), but many early-phase clinical trials are ongoing [136]. AZD9150, an antisense oligonucleotide inhibitor of STAT3, inhibits STAT3 activation in ALCL and nonsmall cell lung cancer lines and demonstrates antitumor activity in lymphoma and lung cancer patient-derived xenograft models [137]. AZD9150 is currently under clinical trials for aggressive non-Hodgkin lymphoma (NCT03527147) and nonsmall cell lung cancer (NCT03334617). Napabucasin (or BBI608), which can inhibit gene expression driven by STAT3 and cancer stemness [138], is being clinically evaluated in solid cancers [139]. 

### 4.2. Upstream Inhibition of STAT3

As STAT3 is downstream to several cytokine and growth factor receptors and their associated JAKs, blocking these receptors by monoclonal antibodies or inhibiting JAKs by small molecular inhibitors represents a promising therapeutic option in both B- and T-cell malignancies. Ruxolitinib has been approved for the treatment of myelofibrosis [140] and polycythemia vera [141]. Ruxolitinib has shown promising antitumor activity in preclinical studies of ABC DLBCL [35], ATLL [142], and MM [117]. Ruxolitinib is currently under a phase 2 clinical study in DLBCL and T-cell non-Hodgkin lymphoma (NCT01431209), a phase 1 trial in MM (NCT03110822), and a clinical investigation in adult T-cell leukemia (ATL) (NCT01712659). Daclizumab, a monoclonal antibody directed to the IL-2 receptor subunit IL-2R, has been shown to be effective in the treatment of patients with chronic ATLL [143]. The therapeutic effectiveness of an anti-IL-6 antibody, Siltuximab (CNTO 328), has been clinically evaluated in Castleman’s disease and MM [144].

### 4.3. Combined Inhibition of JAK/STAT3 and Other Oncogenic Pathways

Because of the crosstalk between the JAK/STAT3 signaling pathway and many other oncogenic pathways (e.g., NF-κB, PI3K, and BCL2) in B- and T-cell lymphoma [65,121], combination therapy that targets JAK/STAT3 and additional oncogenic signaling pathways may increase the effectiveness of treatments. The combination of Ruxolitinib and the BCL-2/BCL-XL inhibitor navitoclax dramatically inhibits growth of primary ATL cells in vitro as well as in an ATL xenograft mouse model [142]. IL-6-induced STAT3 or STAT5 activation is associated with the acquired resistance to PI3K inhibitors in T- and B-cell lymphoma cell lines, and co-targeting JAK/STAT3 and PI3K signaling resensitizes the lymphoma cell lines to a PI3K inhibitor [145].

Our recent studies suggest several potential combination therapeutic strategies in DLBCL [35,70,77]. In ABC DLBCL, the crosstalk between JAK1 and NF-κB signaling promotes cancer cell survival [70]. Co-inhibition of JAK1 and BTK, an early BCR pathway gene that activates NF-κB, by their selective inhibitors synergistically kills ABC DLBCL cells [70]. A phase 1 trial of the JAK1 inhibitor itacitinib and the BTK inhibitor ibrutinib has been initiated in relapsed or refractory DLBCL (NCT02760485). As a transcriptional activator and repressor, STAT3 upregulates genes in multiple oncogenic pathways (e.g., PI3K, cell cycle, and NF-κB) while suppressing gene expression in the lethal type I interferon signaling pathway [35]. Co-treatment of Ruxolitinib and the type I interferon inducer lenalidomide inhibits growth of ABC DLBCL cells in vitro and in vivo [35]. Inhibition of *SGK1*, a target gene of STAT3, by its inhibitor synergizes with AKT inhibitors in the killing of ABC DLBCL cells [77].

## 5. Conclusions

Although STAT3 expression is properly controlled in normal immune responses, constitutive activation of STAT3 is common in human lymphoma. Aberrant activation of STAT3 promotes tumor cell survival, proliferation, and migration through regulating gene expression in multiple oncogenic pathways. In addition, constitutive signaling by STAT3 plays an important role in evasion of immune surveillance. Targeting JAK/STAT3 signaling inhibits tumor growth and enhances antitumor immune responses. Therefore, STAT3 is a promising therapeutic target in lymphoma. Despite significant efforts, no STAT3 inhibitors have yet been approved in the clinic setting. Future perspectives include developing more potent and selective STAT3 inhibitors for cancer therapy.

## Figures and Tables

**Figure 1 cancers-12-00019-f001:**
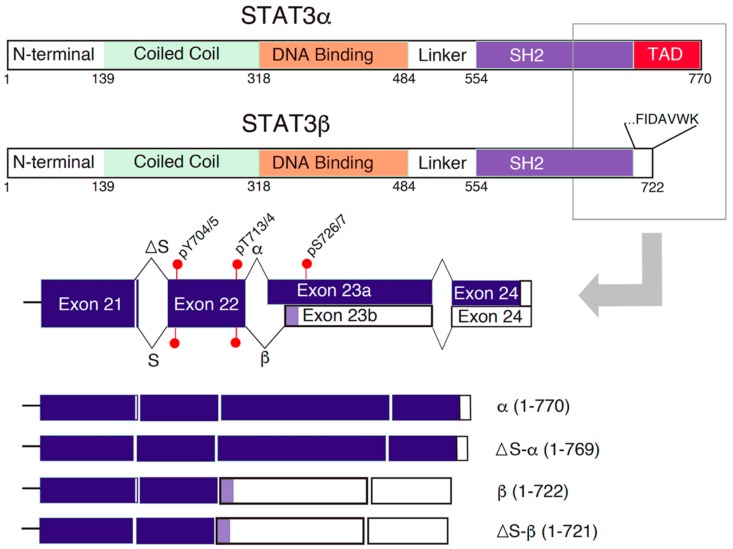
Signal transducer and activator of transcription 3 (STAT3) functional domains and alternative splicing isoforms. Shown are six STAT3 domains: the N-terminus, the coiled-coil domain, the DNA-binding domain, the linker domain, the Src homology 2 (SH2) domain, and the C-terminal transactivation (TAD) domain in STAT3α or unique seven residues in STAT3β. Two splice sites near the 3′ end of the *STAT3* gene produce four STAT3 isoforms: α (the longest isoform with the TAD domain at the C-terminus), β (a shorter isoform with distinct seven residues at the C-terminus), Ser-701-deleted α (ΔS-α), and Ser-701-deleted β (ΔS-β). White boxes indicate the noncoding 3′ UTR, and light blue boxes depict coding sequences due to alternative splicing [20] (see details in the text).

**Figure 2 cancers-12-00019-f002:**
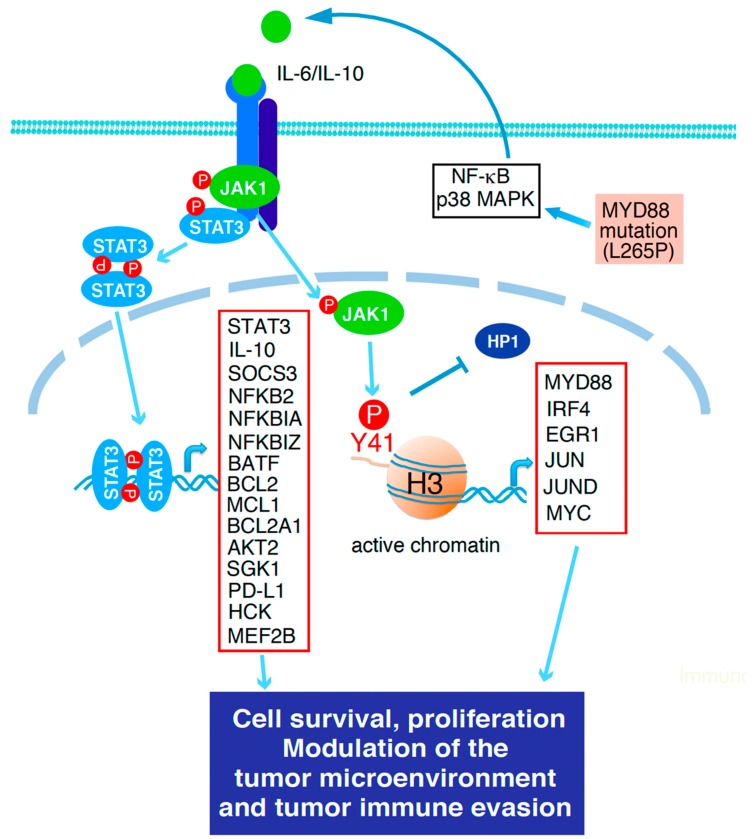
The role of Janus kinase 1 (JAK1)/STAT3 signaling in DLBCL. In ABC DLBCL, JAK1 and STAT3 are constitutively activated by autocrine production of IL-6 and/or IL-10 as a result of MYD88 mutations (mainly L265P) and other NF-κB activation mechanisms. Constitutive activation of JAK1 and STAT3 promotes cell survival and proliferation, modulates the tumor microenvironment and promotes tumor immune evasion through the canonical STAT3 pathway as well as the noncanonical chromatin modification mechanism by JAK1 (see details in the text).

**Figure 3 cancers-12-00019-f003:**
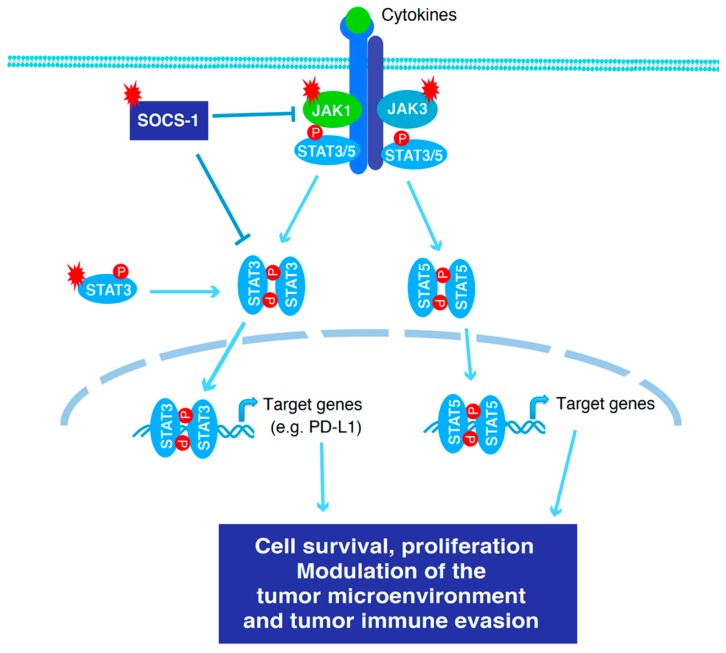
Oncogenic activation of the STAT3 pathway in T-cell lymphoma. Constitutive activation of STAT3 and STAT5, which can be driven by the common gamma chain (γc) receptor-dependent cytokines IL-2, -4, -7, -9, -15, and -21, is pervasive in diverse T-cell malignancies [121]. In PTCL and NKTCL, activating mutations in STAT3, JAK1, and JAK3, and loss of function mutations in SOCS-1 and p53 enhance oncogenic effects of STAT3 and STAT5 by promoting cancer cell survival, modulating the tumor microenvironment and facilitating tumor immune evasion [44].

**Table 1 cancers-12-00019-t001:** STAT3 mutations in lymphoma and related autoimmune diseases.

Location	Disease	Reference
**Gain of function mutations**		
Y640F	T-LGL and NK-LGL, DLBCL, T-cell neoplasm, PTCL, ALCL, ANKL, and NKTCL	[37,38,39,40,41,42,43,44,45,46,47,48,49,50]
D661Y	T-LGL and NK-LGL, T-cell neoplasm, NKTCL, PTCL, ALCL, ITCL, NKTCL, and CLPD-NK	[37,38,39,41,42,43,44,46,47,48,49,50,51]
D661V	T-LGL and CLPD-NK	[37,38,42,46,49,50]
M206K	DLBCL	[52]
K392R	Multisystemic autoimmunity and mycobacterial disease	[53]
M394T	Multisystemic autoimmunity and mycobacterial disease and ALPS	[53,54]
K658N	Multisystemic autoimmunity and mycobacterial disease	[53,55]
R152W	Lymphoproliferation and autoimmunity	[56]
Q344H	Lymphoproliferation, autoimmunity, and PTCL	[45,56]
V353F	Lymphoproliferation and autoimmunity	[56]
E415K	Lymphoproliferation and autoimmunity	[56]
N420K	Lymphoproliferation and autoimmunity	[56]
G421R	Lymphoproliferation and autoimmunity	[56]
T663I	Lymphoproliferation, autoimmunity, and BCLU-DLBCL/B	[40,56]
A703T	Lymphoproliferation and autoimmunity	[56]
T716M	Lymphoproliferation and autoimmunity	[56]
A702T	NKTCL	[43]
G618R	PTCL, NK-LGL, T-LGL, and ANKL	[37,42,43,44,48,55]
S614R	T-LGL, NKTCL, DLBCL and other NHL cases, PTCL, AITL, and ITCL	[37,39,43,44,51,55,57,58]
N647I	T-LGL, NK-LGL, and PTCL	[37,38,44,45,46,49,50,55]
H410R	T-LGL and PTCL	[44,59]
F174S	T-LGL	[23]
D427H	PTCL	[44]
E616G	PTCL, DLBCL, and ITCL	[40,44,51]
E616K	PTCL, DLBCL, and other NHL cases	[44,57]
E696K	PTCL	[44]
R278H	PTCL and ALPS	[44,54]
A662V	ALCL	[47]
**Function to Be Investigated**		
K658M	T-LGL	[37,55]
D661I	NK-LGL	[37]
D661H	T-LGL	[38,50]
Y657_K658insY	T-LGL	[38]
N567K	DLBCL and other NHL cases	[57]
D566N	DLBCL, other NHL cases, and PTCL	[44,57]
R152W	DLBCL	[60]
K658R	DLBCL and T-LGL	[42,46,60]
H447Y	DLBCL	[60]
R278H	DLBCL	[60]
E616del	DLBCL, AITL, and ANKL	[41,48,58,60]
G617R	DLBCL	[40,55]
E616V	T-LGL	[39]
V671F	T-LGL	[39]
S649L	BCLU-DLBCL/B	[40]
Y657ins	T-cell neoplasm and γδ T lymphoma	[41,55]
D661ins	T-cell neoplams	[41]
Y657dup	T-LGL and CLPD-NK	[42,49,55]
K658insY	LGLL	[55]
D171N	PTCL	[44]
V667L	PTCL	[44]
P715L	PTCL	[44]
E638Q	T-LGL	[45]
I659L	T-LGL	[45,49]
K657R	T-LGL	[45]
C550R	AITL	[58]
I659_M660insL	T-LGL	[46]
A662_N663delinsH	T-LGL	[46]
G656_Y657insY	T-LGL	[46]
C614R	NKTCL	[48]
V393A	NKTCL	[48]
H301P	NKTCL	[48]
G656D	PTCL	[61]
Y658M	T-LGL	[49]

Abbreviations: T-LGL: T-cell large granular lymphocytic leukemia; NK-LGL: large granular lymphocytic leukemia of natural killer (NK) cells; LGLL: large granular lymphocytic leukemia; DLBCL: diffuse large B-cell lymphoma; NHL: non-Hodgkin lymphoma; BCLU-DLBCL/B: B-cell lymphoma unclassifiable with features intermediate between diffuse large B-cell lymphoma and Burkitt lymphoma; NKTCL: NK/T-cell lymphomas; PTCL: peripheral T-cell lymphoma; ALPS: autoimmune lymphoproliferative syndrome; ALCL: anaplastic large cell lymphomas; ITCL: intestinal T-cell lymphomas; CLPD-NK: chronic lymphoproliferative disorder of NK cells.

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
