# Peer review of "STAT3 Activation and Oncogenesis in Lymphoma"

_cancers, 2019, doi:10.3390/cancers12010019_

Round 1

Reviewer 1 Report

STAT3, the most studied transcription factor in the Janus kinase (JAK)/STAT signaling pathway, regulates many cellular and biological processes through mediation of various genes’ expression. In this manuscript, authors reported the critical role of STAT3 in lymphoma, and the therapeutic advances for targeting JAK/STAT3 signaling.

The comments are as following:

No STAT3 inhibitors has been approved by FDA to treat cancer yet, please clarify the main obstacle and limitation for these inhibitors. For section 4, much detailed information for each inhibitor should be included, such as clinical trial phase, advantage/disadvantage, side effects and clues for further development. A table or figure including all these described inhibitors would be preferred. . Authors need to provide Figure 3 properly. Figure 3 is totally the same as Figure 2 in the current version. Authors need to revise the whole manuscript carefully for English grammar and type errors. Some special symbols were missing in the current version, such as ‘κ’ for NF-κB, ‘β’ for STAT3β. Multiple myeloma is included in section 3 ‘human lymphoma’, please revise.

Author Response

No STAT3 inhibitors has been approved by FDA to treat cancer yet, please clarify the main obstacle and limitation for these inhibitors. For section 4, much detailed information for each inhibitor should be included, such as clinical trial phase, advantage/disadvantage, side effects and clues for further development. A table or figure including all these described inhibitors would be preferred. . Authors need to provide Figure 3 properly. Figure 3 is totally the same as Figure 2 in the current version. Authors need to revise the whole manuscript carefully for English grammar and type errors. Some special symbols were missing in the current version, such as ‘κ’ for NF-κB, ‘β’ for STAT3β. Multiple myeloma is included in section 3 ‘human lymphoma’, please revise.

Answer: Thanks for the constructive comments. Since some of STAT3 inhibitors are still under clinical trials and not published there is no information detailing side effects and clinical responses. 

We have noticed that editorial formatting of the original files led to some confusion, including changes of greek symbols and misplacing Figure 3 with Figure 2. Please see the revision.

Multiple myeloma is a separate section (3.5). 

Reviewer 2 Report

Some Greek letters were not properly represented. Table 1, footnote was not enough and some of them were incorrect. Figures 2 and 3 were the same. Maybe the figure in Figure 3 was wrong. 

Author Response

Some Greek letters were not properly represented. Table 1, footnote was not enough and some of them were incorrect. Figures 2 and 3 were the same. Maybe the figure in Figure 3 was wrong. 

Answer: Thanks for the comments. These are due to errors by editorial formatting of the original submission. Please see the revised manuscript.

Reviewer 3 Report

[Comments/Suggestions]

This review focuses on the role of STAT3 in lymphoma. However, Abstract is too general, similar with Introduction. I suggest that authors need to summarize the role of STAT3 in lymphoma in Abstract.

For table 1, the information is too little. The authors should elaborate more and make it clearer since not much information can be read in the current format.

Figure 2 and Figure 3 are same figure. They should be combined together.

Typing errors can be found throughout the manuscript, the authors should proofread it carefully.

Author Response

This review focuses on the role of STAT3 in lymphoma. However, Abstract is too general, similar with Introduction. I suggest that authors need to summarize the role of STAT3 in lymphoma in Abstract.

Answer: Thanks for the comments. We have revised the Abstract.

For table 1, the information is too little. The authors should elaborate more and make it clearer since not much information can be read in the current format.

Answer: We have described some mutations in the text. 

Figure 2 and Figure 3 are same figure. They should be combined together.

Typing errors can be found throughout the manuscript, the authors should proofread it carefully.

Answer: The editorial formatting created these errors. Please see the revised manuscript and new Figure 3.

Reviewer 4 Report

The article by Zhu et al., reviews recent findings on STAT3 in lymphoma and provides insight into the oncogenic role of STAT3 in different subtypes of hematopoietic malignancies.

The manuscript covers a reasonable amount of recent literature but it does not allow for careful reviewing in its current form for several reasons:

Formatting throughout the manuscript is incorrect, or incomplete, especially the symbols are missing and there are numerous typing errors in the text. Figure 2 is identical with Figure 3 and therefore no comments can be made on this part of the manuscript. Further, the figures compiled are not easy to follow and do not cover/match what is in the text.

We recommend the authors should submit a more advanced version of their manuscript. The current version of the manuscript is not recommended for publishing.

Author Response

The article by Zhu et al., reviews recent findings on STAT3 in lymphoma and provides insight into the oncogenic role of STAT3 in different subtypes of hematopoietic malignancies.

The manuscript covers a reasonable amount of recent literature but it does not allow for careful reviewing in its current form for several reasons:

Formatting throughout the manuscript is incorrect, or incomplete, especially the symbols are missing and there are numerous typing errors in the text. Figure 2 is identical with Figure 3 and therefore no comments can be made on this part of the manuscript. Further, the figures compiled are not easy to follow and do not cover/match what is in the text.

We recommend the authors should submit a more advanced version of their manuscript. The current version of the manuscript is not recommended for publishing.

Answers; Thanks for the constructive comments. These confusions are largely due to editorial formatting, including changes of greek symbols and misplacing Figure 3 with Figure 2. Please see the revised manuscript and new Figure 3. 

Round 2

Reviewer 1 Report

Authors revised the manuscript and the paper was improved in the current version, while Fig 2 and Fig 3  still look the same.

The manuscript could be considered for publication after minor revision. 

Author Response

Authors revised the manuscript and the paper was improved in the current version, while Fig 2 and Fig 3  still look the same.

The manuscript could be considered for publication after minor revision. 

Answer: Thanks. New Figure 3 has been included. 

Reviewer 4 Report

The review by Zhu et al. has substantially improved with respect to formatting and completeness of the figures.

A general feeling for this review is that the authors put the major focus on their own contributions to the field, which are numerous and relevant. In the more general sections on identification of STAT3 and its signaling properties the review would attain informational content if the authors would acknowledge an additional layer of information or biological context, e.g. quoting the JF Bromberg Cell paper from 1999 not only in the context of own work but also as the hallmark paper showing oncogenic properties of STAT3; quoting that the anti-IFN-I properties of STAT3 in ABC DLBCL originally have been characterized in the antiviral defense u.a.m.

Prior to publication the authors still have to improve style and formatting of the figures:

Fig. 1: the scheme of the splice variants is taken from Turton et al., 2015; this should be quoted in the legend; the legend should be supplemented by information needed when the reader looks at the figure without having the 2015 publication at hand, i.e. abbreviations/symbols (∆SS), amino acid numbering 769/70, 714/5 a.o.m.; the Fig. would also be more comprehensive if separated into upper (A) and lower (B) panel… Fig 2: the scheme implies that IL-6 and IL-10 utilize the same receptor complex and JAK1 is the only receptor-associated JAK family member… the authors should adapt the figure to a more accurate picture of the cytokine signaling events. Fig 3 implies that SOCS1 inhibits JAK3, which according to a recent paper by the Kershaw and Babon groups is not the case (Liau et al Nat Commun 2018); the authors should comment or adapt the figure accordingly.

Finally, the reference section needs careful revision with respect to completeness and style of the quotations.

Author Response

general feeling for this review is that the authors put the major focus on their own contributions to the field, which are numerous and relevant. In the more general sections on identification of STAT3 and its signaling properties the review would attain informational content if the authors would acknowledge an additional layer of information or biological context, e.g. quoting the JF Bromberg Cell paper from 1999 not only in the context of own work but also as the hallmark paper showing oncogenic properties of STAT3; quoting that the anti-IFN-I properties of STAT3 in ABC DLBCL originally have been characterized in the antiviral defense u.a.m.

Answer: Thanks for insightful comments. We have revised accordingly.

Prior to publication the authors still have to improve style and formatting of the figures:

Fig. 1: the scheme of the splice variants is taken from Turton et al., 2015; this should be quoted in the legend; the legend should be supplemented by information needed when the reader looks at the figure without having the 2015 publication at hand, i.e. abbreviations/symbols (∆SS), amino acid numbering 769/70, 714/5 a.o.m.; the Fig. would also be more comprehensive if separated into upper (A) and lower (B) panel… Fig 2: the scheme implies that IL-6 and IL-10 utilize the same receptor complex and JAK1 is the only receptor-associated JAK family member… the authors should adapt the figure to a more accurate picture of the cytokine signaling events. Fig 3 implies that SOCS1 inhibits JAK3, which according to a recent paper by the Kershaw and Babon groups is not the case (Liau et al Nat Commun 2018); the authors should comment or adapt the figure accordingly.

Answer: We have modified Figures 1 and 3 based on reviewer's important suggestions. In Figure 2, JAK1 is the only active kinase that mediates IL6/IL10 signaling. JAK2 has been reported by a study but our data do not support this study.

Finally, the reference section needs careful revision with respect to completeness and style of the quotations.

Answer: revised.